# Aortopathies in mouse models of Pompe, Fabry and Mucopolysaccharidosis IIIB lysosomal storage diseases

Maria Paola Belfiore[1◦], Francesca Iacobellis[1◦], Emma Acampora[2], Martina Caiazza[3], Marta Rubino[3], Emanuele Monda[3], Maria Rosaria Magaldi[3], Antonietta Tarallo[2], Marcella Sasso[2], Valeria De Pasquale[4], Roberto Grassi[1], Salvatore Cappabianca[1], Paolo Calabrò[3], Simona Fecarotta[2], Salvatore Esposito[5], Giovanni Esposito[2], Antonio Pisani[2], Luigi Michele Pavone[4], Giancarlo Parenti[2‡], Giuseppe Limongelli[3,6‡]*

1 Department of Radiology, University of Campania "L. Vanvitelli", Naples, Italy, 2 Department of Translational Medical Sciences, Federico II University, Naples, Italy, 3 Department of Translational Medical Sciences, University of Campania "Luigi Vanvitelli", Naples, Italy, 4 Department of Molecular Medicine and Medical Biotechnology, Federico II University, Naples, Italy, 5 Unit of Pathological Anatomy, Aversa Hospital, Caserta, Italy, 6 Institute of Cardiovascular Sciences, University College of London and St. Bartholomew's Hospital, London, United Kingdom

◦ These authors contributed equally to this work.
‡ These authors share last authorship on this work.
* limongelligiuseppe@libero.it

**Data Availability Statement:** All relevant data are within the paper and its Supporting Information files.

## Abstract

### Introduction

Lysosomal storage diseases (LSDs) are rare inherited metabolic diseases characterized by an abnormal accumulation of various toxic materials in the cells as a result of enzyme deficiencies leading to tissue and organ damage. Among clinical manifestations, cardiac diseases are particularly important in Pompe glycogen storage diseases (PD), in glycosphingolipidosis Fabry disease (FD), and mucopolysaccharidoses (MPS). Here, we evaluated the occurrence of aortopathy in knock out (KO) mouse models of three different LSDs, including PD, FD, and MPS IIIB.

### Methods

We measured the aortic diameters in 15 KO male mice, 5 for each LSD: 5 GLA$^{-/-}$ mice for FD, 5 NAGLU$^{-/-}$ mice for MPS IIIB, 5 GAA$^{-/-}$ mice for PD, and 15 wild type (WT) mice: 5 for each strain. In order to compare the aortic parameters between KO and WT mice deriving from the same colonies, different diameters were echocardiographically measured: aortic annulus, aortic sinus, sino-tubular junction, ascending aorta, aortic arch and descending aorta. Storage material content and aortic defects of the KO mice were also analyzed by histology, when available.

### Results

Compared to their correspondent WT mice: GAA$^{-/-}$ mice showed greater diameters of ascending aorta (1.61mm vs. 1.11mm, p-value = 0.01) and descending aorta (1.17mm vs

**Funding:** The author(s) received no specific funding for this work.

**Competing interests:** The authors have declared that no competing interests exist.

1.02mm, p-value 0.04); GLA$^{-/-}$ mice showed greater diameters of aortic annulus (1.35mm vs. 1.22mm, p-value = 0.01), sinus of Valsalva (1.6mm vs. 1.38mm, p-value<0.01), ascending aorta (1.57mm vs. 1.34mm, p-value<0.01), aortic arch (1.36mm vs. 1.22mm, p-value = 0.03) and descending aorta (1.29mm vs. 1.11mm, p-value<0.01); NAGLU$^{-/-}$ mice showed greater diameters of sinus of Valsalva (1.46mm vs. 1.31mm, p-value = 0.05), ascending aorta (1.42mm vs. 1.29mm, p-value<0.01), aortic arch (1.34mm vs. 1.28mm, p-value<0.01) and descending aorta (1.18mm vs. 1.1mm, p-value 0.01).

## Conclusions

We evaluated for the first time the aortic diameters in 3 LSD mouse models and identified different aortopathy patterns, in concordance with recent human findings. Our results are relevant in view of using KO mouse models for efficiently testing the efficacy of new therapies on distinct cardiovascular aspects of LSDs.

## Introduction

Lysosomal storage diseases (LSDs) are rare inherited metabolic diseases due to defects of lysosomal functions, in most cases, caused by deficiencies of acidic hydrolases involved in the degradation of complex molecules [1,2]. Consequently, in these disorders, storage material accumulates in the lysosomal compartment of the cells, leading to altered cellular processes with variable involvement of multiple tissues and organs [3,4]. Thus, LSDs are multisystem disorders with broad phenotypes that range from early-onset severe forms to late-onset forms characterized by attenuated clinical course, and milder manifestations [1]. Among the clinical symptoms in LSDs, cardiac dysfunction is particularly relevant in lysosomal glycogen storage diseases (Pompe and Danon disease), glycosphingolipidoses (Anderson-Fabry disease), and mucopolysaccharidoses [4]. Hypertrophic and dilated cardiomyopathy, coronary artery disease, and valvular diseases are the most common disease manifestations in affected patients. In this study, we focused on the occurrence of aortopathy in the mouse models of three LSDs: Pompe disease (PD), Fabry disease (FD), and Mucopolysaccharidosis (MPS) IIIB.

PD is a rare autosomal recessive and progressive genetic disorder caused by mutations in the gene encoding for the acid α-glucosidase (GAA), the lysosomal enzyme that degrades glycogen [5]. The clinical spectrum of the disease is broad, ranging from the infantile-onset PD, associated with glycogen accumulation in heart and muscles, leading to premature death, if not treated [6–8], to the late-onset PD, with predominant skeletal muscle and vascular involvement, including the ascending aorta [9,10].

FD is an X-linked recessive LSD due to mutations in the gene encoding for the α-galactosidase (GLA), leading to the accumulation of globotriaosylceramide in the cells of various tissues with con-sequent multi-organ dysfunction [11,12]. Clinical manifestations of FD include systemic vasculopathy resulting in a markedly increased risk of ischemic stroke, small-fiber peripheral neuropathy, cardiac dysfunction, and chronic kidney disease [13–15].

MPSs are inborn errors of metabolism due to the deficiency of lysosomal enzymes involved in the degradation of glycosaminoglycans (GAGs) [16]. Clinical manifestations of MPSs include neurological disorders as well as skeletal, joint, airway and cardiac defects, hearing and vision impairment, and mental retardation. The affected patients usually die in the second or third decade of their life. Depending on the accumulated GAGs, MPSs are classified into seven

types (I, II, III, IV, VI, VII, and IX) that are variable in their prevalence, clinical symptoms, and degree of severity. In MPS patient heart, accumulation of GAGs occurs within the cardiac valves, the epicardial coronary arteries, the myocytes, the cardiac interstitium and the walls of the great vessels, thus triggering cardiac valve regurgitation and stenosis, diffuse coronary artery stenosis, myocardial dysfunction and aortic root dilation [17–21]. Cardiac disease is highly prevalent (60%-100% of patients) in MPS type I, II, and VI affected patients [22]. However, evidence exists on cardiac involvement in MPS III as well. Indeed, valvulopathies, mainly involving the mitral and aortic valves, cardiomyopathy, arrhythmias, coronary artery disease, aortic root dilatation, and conduction abnormalities have been reported for MPS III affected patients [17–18,23–24]. The subtype MPS IIIB is due to the deficiency of N-acetyl-α-glucosa-minidase (NAGLU), which is required for the degradation of heparan sulfate (HS). We recently demonstrated that heart disease, valvular abnormalities, and cardiac failure is associated with an impaired lysosomal autophagic flux in the mouse model of MPS IIIB [25].

Enzyme replacement therapy (ERT) substantially improves many of the features of the above described LSDs, including some aspects of the cardiac involvement [3,5,26]. However, several disease-modifying treatment options, including chaperone-based therapy, mRNAs, and gene therapies, are currently under investigation [3–5,26–31]. Thus, extensive knowledge of the prevalence and nature of cardiac defects in these LSDs may be relevant. In particular, the presence and characteristics of aortopathy related to LSDs have not been investigated in pre-clinical models of LSDs. This study sought to study aortic size and morphology in knock out (KO) mouse models of 3 different LSDs: PD, FD, and MPS IIIB.

## Materials and methods

### Animals

Animal studies were performed according to the EU Directive 86/609 regarding the protection of animals used for experimental purposes, and according to Institution Animal Care and use committee (IACUC) guidelines for the Care and Use of animals in research. The study was approved by the Italian Ministry of Health, IACUC n. 523/2015-PR (06/11/2015). Every procedure on the mice was performed to ensure that discomfort, distress, pain, and injury would be minimal.

Thirty mice were examined: 15 were KO male mice for the LSDs: 5 GLA$^{-/-}$, 5 NAGLU$^{-/-}$, 5 GAA$^{-/-}$; the remaining were 15 wild type (WT) mice: 5 for each strain. All of them were examined at 12 months. They were maintained on light/dark cycles of 12/12 h and had free access to food and water.

The NAGLU$^{-/-}$ murine model of MPS IIIB was previously generated by the insertion of the neomycin resistance gene into exon 6 of the NALGU gene on the C57/BL6 background. NAGLU$^{-/-}$ and wildtype (WT) mice derived from the same colony were genotyped by PCR [32].

The GAA$^{-/-}$ murine model of PD was previously created by insertion of the neomycin resistance gene and the herpes virus thymidine kinase gene in the locus encoding GAA. Six independent cell lines containing the disrupted GAA allele were used to make chimeras that were bred to C57/BL6 females to generate heterozygous mice (F1). GAA$^{-/-}$ and wildtype (WT) mice derived from the same colony were genotyped by PCR [33].

The GLA$^{-/-}$ murine model of FD was previously generated by a targeted disruption of the α-GLA gene on the C57/BL6 background. GLA$^{-/-}$ and wildtype (WT) mice derived from the same colony were genotyped by PCR [34].

## Ultrasound analysis

M-mode, two-dimensional, Color Doppler echocardiography was performed by a high-resolution ultrasound machine (VisualSonics Vevo 2100 unit, Fujifilm) capable of recording over 750 frames per second and equipped with a high-frequency (40 MHz) transducer. During each imaging session, mice were anesthetized by inhalation with isoflurane in oxygen (3–4% for induction and 1.5% for maintenance).

Mice were allowed to breathe spontaneously. Heart rate and body temperature were appropriately monitored by cardiac electrodes integrated in the heating pad and by a rectal probe and remained constant throughout the examination. In order to prevent imaging artifacts, each mouse was immobilized, and its chest hair was removed, applying a calcium thioglycolate depilatory cream. The transducer was fixed on a special arm. An appropriate amount of heated sonographic gel (Aquasonic 100; Parker Laboratories, Inc, Fairfield, NJ) was applied to the shaved skin.

The transducer was positioned longitudinally to its body, along the right parasternal line, with the index marker pointing to its head and rotated about 35° counterclockwise. The obtained B-mode image represents a section across the long axis of the left ventricle, which allowed visualization of the aortic root and the outflow tract of the left ventricle.

The following echo views were examined: parasternal long axis; parasternal short axis; apical; aortic arch. The last was obtained from a modified right parasternal view, with the transducer positioned longitudinally to the mouse body, along the right parasternal line, almost parallel to the table and with the index marker pointing to the head. This view is optimal to morphologically examine the ascending aorta and the arch, as well as the main arterial branches. Measured diameters were aortic annulus, aortic sinus, sino-tubular junction, ascending aorta, aortic arch, and descending aorta. Annulus dimensions were obtained in the parasternal long-axis view during systole for semilunar valves, and the apical four-chamber view during diastole for atrioventricular valves as previously described [25,35].

## Histochemical analysis

Immediately after echocardiographic imaging, mice (n = 5 NAGLU[−/−] and n = 5 correspondent WT from the same colony, n = 5 GAA[−/−] and n = 5 correspondent WT from the same colony) were euthanized by cervical dislocation in compliance with the recommendations contained in the American Veterinary Medical Association (AVMA) Guidelines for the Euthanasia of Animals.

Subsequently, hearts were harvested for histological and morphometric analyses. Hearts were isolated, fixed in a bath of 4% aqueous buffered formalin overnight, processed for paraffin embedding [36,37], and coronal sections (10 μm thick), containing right and left ventricles, and aorta, were obtained. Sequential sections from each heart were stained with hematoxylin and eosin (H&E) staining (from Sigma-Aldrich) [38], and periodic acid-Schiff (PAS)-Alcian Blue stain (from Dako) for storage content evaluation. Selected sections were stained with Weiger for elastic fiber evaluation. Image analysis was performed with a charge-coupled device camera coupled with a light microscope.

## Statistical analysis

In the post-processing phase, the acquired ultrasound data were exported and analyzed using the dedicated software (Vevo 2100, Visual Sonics). Statistical analyses were performed using SPSS (version 25.0, SPSS Inc., Chicago, IL, USA). Normally distributed continuous data are presented as mean ± standard deviation (SD) and were compared by t-test or ANOVA (if more than two groups). Values of $p < 0.05$ were considered statistically significant.

**Table 1. Comparison between diameters of the different portions of the aorta of GAA knock-out mice (GAA⁻/⁻) and their correspondent wild-type (WT) mice.**

|  | GAA⁻/⁻ (Pompe) (n = 5) | WT (n = 5) | p-value |
|---|---|---|---|
| Aortic annulus | 1.51 (±0.22) | 1.34 (±1.11) | 0.311 |
| Sinus of Valsalva | 1.88 (±0.26) | 1.6 (±0.08) | 0.108 |
| Sino-tubular junction | 1.49 (±0.23) | 1.26 (±0.07) | 0.100 |
| Ascending aorta | 1.61 (±0.17) | 1.11 (±0.03) | 0.010* |
| Aortic arch | 1.51 (±0.24) | 1.17 (±0.04) | 0.087 |
| Descending aorta | 1.17 (±0.09) | 1.02 (±0.09) | 0.036* |

## Results

The weight of the KO mice for each LSD model was not significantly different compared to their correspondent WT: the medium weight of GLA⁻/⁻ and of their correspondent WT was 29.2 ±2.2 and 28.7 ±1.2, respectively; the medium weight of NAGLU⁻/⁻ and of their correspondent WT was 28.9 ±0.7 and 30 ±1.0, respectively; the medium weight of GAA⁻/⁻ and of their correspondent WT was 28 ±2.0 and 28.2 ±1.0, respectively.

In order to evaluate the aortic parameters in the three different LSD mouse models, the echocardiographic measurement of different portions of the aorta (aortic annulus, aortic sinus, sino-tubular junction ascending aorta, aortic arch and descending aorta) was performed (Tables 1–3).

In particular, compared to their correspondent WT mice:

GAA KO mice (GAA⁻/⁻) showed greater diameters of ascending aorta (1.61mm vs. 1.11mm, p-value = 0.01) and descending aorta (1.17mm vs. 1.02mm, p-value = 0.04), while no difference statistically significant was reported in the other aortic parameters evaluated (Table 1);

**Table 2. Comparison between diameters of the different portions of the aorta of GLA knock-out mice (GLA⁻/⁻) and their correspondent wild-type (WT) mice.**

|  | GLA⁻/⁻ (Fabry) (n = 5) | WT (n = 5) | p-value |
|---|---|---|---|
| Aortic anulus | 1.35 (±0–44) | 1.22 (±0.01) | 0.018* |
| Sinus of Valsalva | 1.6 (±0.05) | 1.38 (±0.03) | <0.001* |
| Sino-tubular junction | 1.26 (±0.05) | 1.22 (±0.05) | 0.703 |
| Ascending aorta | 1.57 (±0.04) | 1.34 (±0.06) | <0.001* |
| Aortic arch | 1.36 (±0.03) | 1.22 (±0.07) | 0.030* |
| Descending aorta | 1.29 (±0.07) | 1.11 (±0.04) | <0.001 |

**Table 3. Comparison between diameters of the different portions of the aorta of NAGLU knock-out mice (NAGLU⁻/⁻) and their correspondent wild-type (WT) mice.** MPS: Mucopolysaccharidosis.

|  | NAGLU⁻/⁻ (MPS IIIB) (n = 5) | WT (n = 5) | p-value |
|---|---|---|---|
| Aortic annulus | 1.36 (±0.09) | 1.26 (±0.1) | 0.060 |
| Sinus of Valsalva | 1.46 (±0.11) | 1.31 (±0.03) | 0.049* |
| Sino-tubular junction | 1.16 (±0.08) | 1.13 (±0.04) | 0.536 |
| Ascending aorta | 1.42 (±0.05) | 1.29 (±0.06) | 0.005* |
| Aortic arch | 1.34 (±0.04) | 1.28 (±0.04) | 0.002* |
| Descending aorta | 1.18 (±0.04) | 1.1 (±0.04) | 0.012* |

**Table 4. Comparison between diameters of the different portion of aorta in knock-out mouse models of the three different lysosomal storage disease.** MPS: Mucopolysaccharidosis.

| | GAA $^{-/-}$ (Pompe) (n = 5) | GLA $^{-/-}$ (Fabry) (n = 5) | NAGLU $^{-/-}$ (MPS IIIB) (n = 5) | p-value |
|---|---|---|---|---|
| Aortic annulus | 1.51 (±0.22) | 1.35 (±0.44) | 1.36 (±0.09) | 0.176 |
| Sinus of Valsalva | 1.88 (±0.26) | 1.6 (±0.05) | 1.46 (±0.11) | 0.006* |
| Sino-tubular junction | 1.49 (±0.23) | 1.26 (±0.05) | 1.16 (±0.08) | 0.007* |
| Ascending aorta | 1.61 (±0.17) | 1.57 (±0.04) | 1.42 (±0.05) | 0.037* |
| Aortic arch | 1.51 (±0.24) | 1.36 (±0.03) | 1.34 (±0.04) | 0.149 |
| Descending aorta | 1.17 (±0.09) | 1.29 (±0.07) | 1.18 (±0.04) | 0.028* |

GLA KO mice (GLA$^{-/-}$) showed greater diameters of aortic annulus (1.35mm vs. 1.22mm, p-value = 0.01), sinus of Valsalva (1.6mm vs. 1.38mm, p-value<0.01), ascending aorta (1.57mm vs. 1.34mm, p-value<0.01), aortic arch (1.36mm vs. 1.22mm, p-value = 0.03) and descending aorta (1.29mm vs. 1.11mm, p-value<0.01), while no difference statistically significant was reported considering the sino-tubular junction (Table 2);

NAGLU KO mice (NAGLU$^{-/-}$) showed greater diameters of sinus of Valsalva (1.46mm vs. 1.31mm, p-value = 0.05), ascending aorta (1.42mm vs. 1.29mm, p-value<0.01), aortic arch (1.34mm vs. 1.28mm, p-value<0.01) and descending aorta (1.18mm vs. 1.1mm, p-value = 0.01). At the same time, no difference statistically significant was reported considering the aortic annulus and the sino-tubular junction (Table 3).

Moreover, a comparison of the echocardiographic measurements between the 3 LSD KO mouse models was also performed and reported in Table 4.

GAA KO mice (GAA$^{-/-}$) showed greater diameters compared to GLA KO mice (GLA$^{-/-}$) and NAGLU KO mice (NAGLU$^{-/-}$) considering aortic sinus (1.88mm, 1.6mm e 1.46mm, respectively, p-value<0.01), the sino-tubular junction (1.49mm, 1.26mm e 1.16mm, respectively, p-value<0.01) and ascending aorta (1.61mm, 1.57mm e 1,42mm, respectively, p-value = 0.04).

GLA KO mice (GLA$^{-/-}$) showed greater diameters compared to GAA KO mice (GAA$^{-/-}$) and NAGLU KO mice (NAGLU$^{-/-}$) considering descending aorta (1.29mm, 1.17mm e 1.18mm, respectively, p-value = 0.03); no difference statistically significant was reported considering the aortic annulus and the aortic arch. Moreover, compared to their correspondent WT and other KO models, NAGLU$^{-/-}$ mice exhibited increased thickening of the aortic valve leaflets.

To evaluate aortic structures in the analyzed LSD mouse models (NAGLU$^{-/-}$ and GAA$^{-/-}$), we performed hematoxylin-eosin (H&E) and Alcian blue staining of heart sections. According to the aortic defects detected by echocardiography at the time point examined, H&E and PAS staining of sections from the hearts of GAA$^{-/-}$ mice evidenced an aortic wall with mild disorganization of lamellar units for the presence of many vacuoles containing fine granular or amorphous material in their reduced inter-lamellar space (Fig 1). The aortic valve was not thickened, nor accumulation was found. H&E and PAS staining of sections from the hearts of NAGLU$^{-/-}$ mice confirmed the aortic valve defects detected by echocardiography at the time point examined. NAGLU$^{-/-}$ mice exhibited significant aortic cuspid thickening with excess/redundant tissue present in the valves, as demonstrated by Alcian blue-PAS staining (Fig 2). By using the H&E staining, NAGLU$^{-/-}$ aortic valve cuspids exhibited a disruption of the normal collagen/proteoglycan boundary interfaces observed in WT animals (Fig 2).

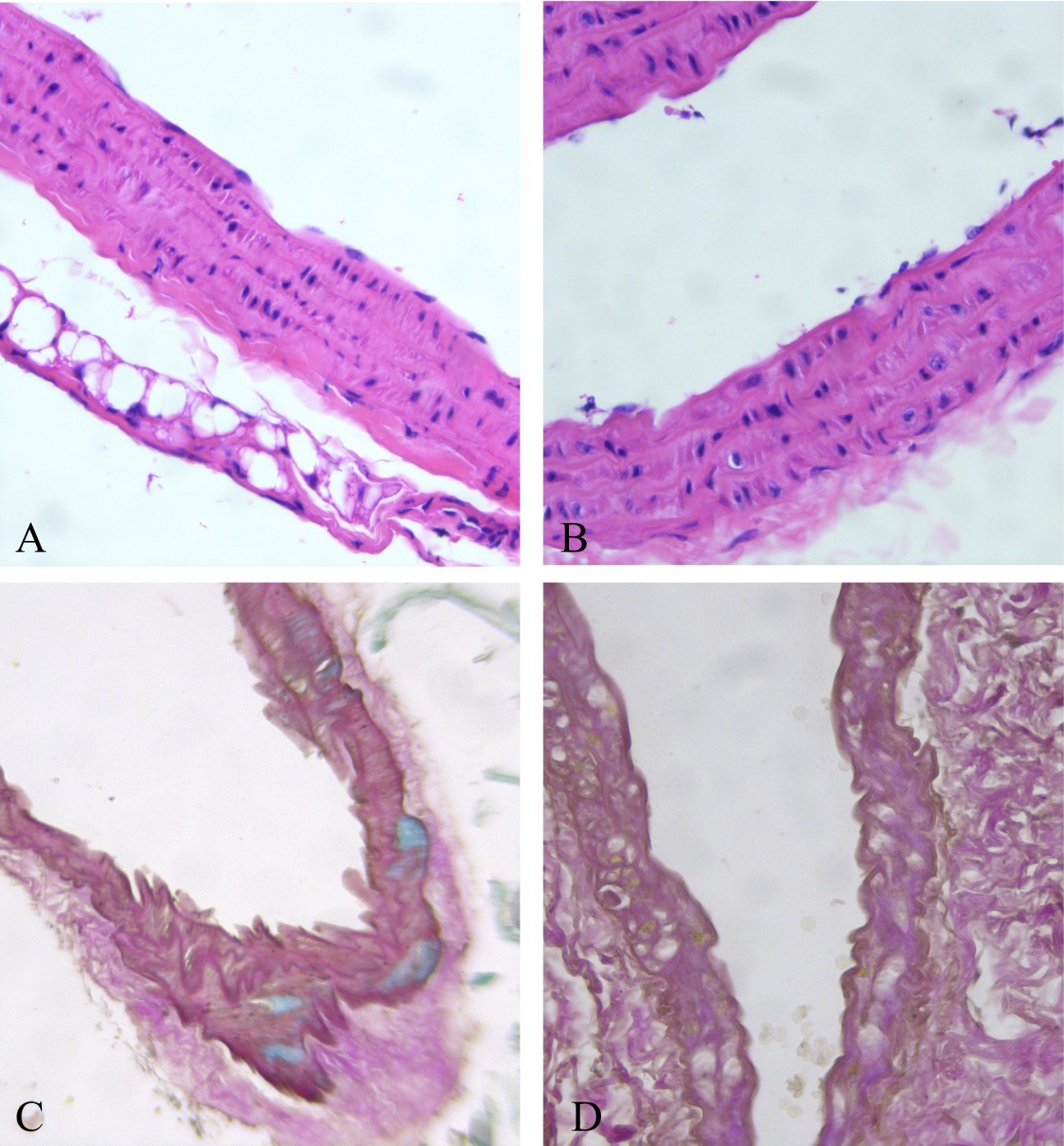

**Fig 1. Large empty vacuoles in the external third of the aortic wall in the GAA$^{-/-}$ mouse heart (A and B, H&E 40x), causing lamellar unit disorganization (C, Alcian-Weighert 20x, and D, PAS-Weighert 40x).**

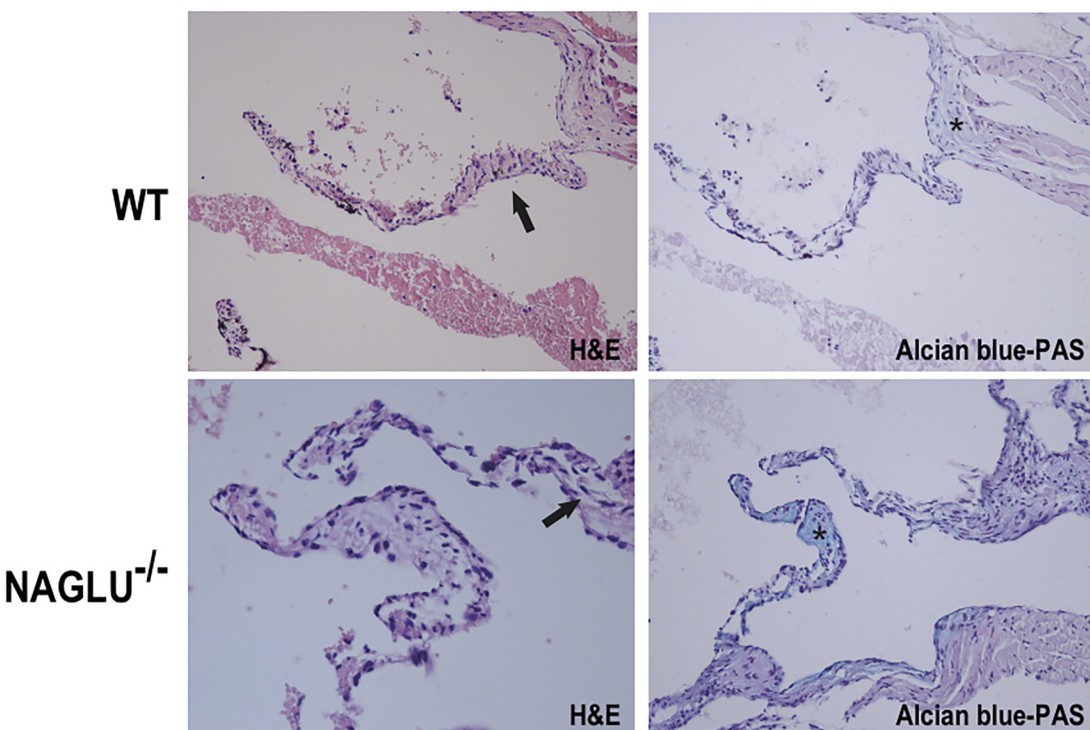

**Fig 2. Aortic valve morphology alterations in the MPS IIIB mouse model.** Representative images of hematoxylin-eosin (H&E) (left) and Alcian blue-PAS (right) staining of aortic valve sections from WT and NAGLU[-/-] hearts.

## Discussion

In literature, inherited aortopathies are well known to be associated with genetic syndromes (e.g., Marfan syndrome) and collagen disease (e.g., Ehlers Danlos), or with familial diseases (e.g., bicuspid aortic valve aortopathy) [15]. Cardiac involvement in LSD generally involves the heart muscle and cardiac valves [14–18]. However, a growing field of interest is the study of the aortic size in LSDs [8–11]. The concept of "metabolic storage aortopathies" is just emerging with the evidence that mild to moderate aortic enlargement, and few cases of aneurysm or dissection have been reported in LSDs. To date, it is not clear whether aortic dilation is an LSD general feature, or it is typical of specific disorders.

To our knowledge, this is the first report of aortopathy in mouse models of LSDs. Besides, different patterns of dilation have been associated with different models and compared to human findings from literature. In our study, we found a difference between PD vs. WT mice for ascending and descending aorta parameters. This finding is consistent with the data reported in the literature. El-Gharbawy et al. [9] reported dilated arteriopathy involving the ascending thoracic aorta primarily in 5 female patients with late-onset PD, including one patient with bicuspid aortic valve, who developed dissection; another patient with juvenile-onset disease showed both thoracic and basilar artery aneurysms. In the present study, histochemical evaluation in GAA[-/-] mice evidenced signs of disorganization and storage in the aortic wall. This finding can probably explain the increased aortic stiffness identified in patients with PD [39].

In the FD model, aortic dilation was diffuse (except for sino-tubular junction), and particularly evident at aortic sinus and ascending aorta. Barbey et al. [14], studying a cohort of 106 patients with Fabry disease, showed aortic dilation at the sinus of Valsalva in 32.7% of males and 5.6% of females; aneurysms were present in 9.6% of males and 1.9% of females.

In the MPS IIIB mouse model, except for sino-tubular junction, aortic dilation was diffuse; a non-significant trend was found at the annulus. Moreover, a thickening of the aortic valve was evidenced and confirmed by histochemical analysis. This analysis shows an excess/redundant tissue present in the valves (demonstrated by Alcian blue-PAS staining) and disruption of the regular collagen/proteoglycan boundary interfaces observed in WT animals (H&E staining). In a cohort of MPS I-VII patients (age range: 3.4–25.9 years), about one-third developed aortic dilation, with the highest prevalence in MPS IVA (87.5%). Aortic dilation was prevalent at the annulus (41%; 14/34) and sinus of Valsalva (35%; 12/34) [19]. Poswar et al. [20] studied 69 patients with MPS, showing a prevalence of up to 39%, particularly in MPS IVa and MPS VI. No significant effect of enzyme replacement therapy (ERT) on aortic size was found in 11 patients with available data (2 with MPS I; 4 with MPS II; 2 with MPS IVA, and 3 with MPS VI). The pathogenesis is not clear. However, in MPS VII mice, GAG accumulation seems to activate complement components, which may play a role in signal transduction pathways that upregulate elastases [40]. In MPS canine models, interleukin 6-like cytokine oncostatin M resulted to be increased in the aorta of MPS I and MPS VII dogs, and tumor necrosis factor-α and toll-like receptor 4 were increased in MPS VII dog aortas, suggesting that these cytokines could contribute to the upregulation of the elastases [41]. Similarly, in the NAGLU$^{-/-}$ mouse model also the aortic valve was involved, showing an increased thickening of the aortic valve leaflets compared with WT.

Here, we studied three KO mouse models, which recapitulate LSDs that differ in terms of enzymatic defects, stored substrates (glycogen in PD, glycosphingolipids in FD, and glycosaminoglycans in MPS IIIB), and clinical manifestations. Despite these differences, in all animal models, we found signs of aortic involvement, suggesting that this type of manifestations are common findings in LSDs, and should be carefully looked at in affected patients. Interestingly, ascending aorta dilatation was significantly more relevant in PD mice compared to the other two LSD models. On the other hand, FD mice showed a significant dilation at the descending aorta compared with the other two LSD models. This is a new finding since no data are reported in the literature regarding FD affected patients.

## Study limitations

Our study has some limitations, including the small sample size and the single time for the aortic evaluation. Unfortunately, we were not able to have histochemical findings of FD mice (due to technical problems). Moreover, this pilot study was essentially designed to describe the aortic phenotype in different mouse models of lysosomal storage disease. Thus, based on these results, the effect of old and new treatments (ERT, losartan, genetic therapies) on our models will be the objective of future investigations.

## Conclusions

In conclusion, we evaluated for the first-time aortic diameters in 3 LSD mouse models and identified different aortopathy patterns, in concordance with recent human findings. Our results are relevant in view of using KO mouse models for efficiently testing the efficacy of new therapies on distinct cardiovascular aspects of LSDs.

## Supporting information

**S1 Checklist. The ARRIVE guidelines checklist.**
(PDF)

## Acknowledgments

Author Giuseppe Limongelli is a member of ERN GUARD-HEART (European Reference Network for Rare and Complex Diseases of the Heart; http://guardheart.ern-net.eu).

## Author Contributions

**Conceptualization:** Marta Rubino.

**Data curation:** Marta Rubino, Roberto Grassi, Salvatore Cappabianca.

**Formal analysis:** Martina Caiazza.

**Investigation:** Emma Acampora, Martina Caiazza.

**Methodology:** Emma Acampora, Salvatore Cappabianca.

**Project administration:** Giovanni Esposito.

**Resources:** Giovanni Esposito.

**Supervision:** Paolo Calabrò, Simona Fecarotta, Antonio Pisani, Luigi Michele Pavone.

**Visualization:** Luigi Michele Pavone.

**Writing – original draft:** Maria Paola Belfiore, Francesca Iacobellis, Emanuele Monda, Maria Rosaria Magaldi, Antonietta Tarallo, Marcella Sasso, Antonio Pisani, Luigi Michele Pavone, Giancarlo Parenti, Giuseppe Limongelli.

**Writing – review & editing:** Maria Paola Belfiore, Francesca Iacobellis, Emanuele Monda, Valeria De Pasquale, Salvatore Esposito, Luigi Michele Pavone, Giancarlo Parenti, Giuseppe Limongelli.

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
