## [Decision Letter · Decision Letter 0]

17 Jan 2020

PONE-D-19-29826

Lysosomal Storage Disease: Pompe, Fabry and Mucopolysaccharidoses. The emerging concept of lysosomal aortopathies

PLOS ONE

Dear Prof., Dr., Limongelli,

Thank you for submitting your manuscript to PLOS ONE. After careful consideration, we feel that it has merit but does not fully meet PLOS ONE’s publication criteria as it currently stands. Therefore, we invite you to submit a revised version of the manuscript that addresses the points raised during the review process.

 Please respond to all points raised by the reviewers. 

Additional experiments must be done to render the manuscript suitable for publication.

In particular, since data on the analyzed parameter are already available in human patients, other parameters relevant for the observed aortopathie and not easily measured in human samples must be assessed. 

As suggested by both reviewers storage material/lysosomal pathology should be investigated in order to provide some mechanistic explanation to the observed data. 

Data showing the response of all investigated parameters to therapy must also be provided. 

We would appreciate receiving your revised manuscript by Mar 02 2020 11:59PM. To enhance the reproducibility of your results, we recommend that if applicable you deposit your laboratory protocols in protocols.io, where a protocol can be assigned its own identifier (DOI) such that it can be cited independently in the future. For instructions see: http://journals.plos.org/plosone/s/submission-guidelines#loc-laboratory-protocols

We look forward to receiving your revised manuscript.

Kind regards,

Andrea Dardis, Ph.D.

Academic Editor

PLOS ONE

Journal Requirements:

2. As part of your revision, please complete and submit a copy of the ARRIVE Guidelines checklist, a document that aims to improve experimental reporting and reproducibility of animal studies for purposes of post-publication data analysis and reproducibility: https://www.nc3rs.org.uk/arrive-guidelines. Please also add method of sacrifice and source of animals in the Methods section of your manuscript. Please include your completed checklist as a Supporting Information file. Note that if your paper is accepted for publication, this checklist will be published as part of your article.

3. We noticed you have some minor occurrence(s) of overlapping text with the following previous publication(s), which needs to be addressed:

https://doi.org/10.1038/s41436-018-0103-8

https://doi.org/10.1186/1476-7120-9-39

In your revision ensure you cite all your sources (including your own works), and quote or rephrase any duplicated text outside the Methods section. Further consideration is dependent on these concerns being addressed.

Reviewers' comments:

Reviewer's Responses to Questions

**Comments to the Author**

1. Is the manuscript technically sound, and do the data support the conclusions?

Reviewer #1: Yes

Reviewer #2: No

2. Has the statistical analysis been performed appropriately and rigorously? 

Reviewer #1: I Don't Know

Reviewer #2: Yes

3. Have the authors made all data underlying the findings in their manuscript fully available?

Reviewer #1: Yes

Reviewer #2: Yes

4. Is the manuscript presented in an intelligible fashion and written in standard English?

Reviewer #1: Yes

Reviewer #2: Yes

5. Review Comments to the Author

Reviewer #1: This is an interesting manuscript that highlights an aspect of LSDs that is not often considered. I wonder whether the authors can provide a little more data on parameters that are relevant for the aortopathies that they observe? For example cell biological analysis of storage material/lysosomal pathology? Other minor comments are listed below.

- Literature citations and discussion are incomplete, no mentioning has been made on aortic stiffness in human patients. These are relevant to this manuscript.

- The introduction is too short in my opinion. The question: 'aortopathies…have not been investigated in pre-clinical models…' is not well explained. Why is this study relevant? It is in my opinion, but please make this more clear and provide a more comprehensive reasoning to introduce the topic

- please provide information on the genetic background of the mice and corresponding wild type mice.

- Please provide the data for the weight of the ko mouse models at analysis

- The figures are redundant with the tables and could be removed

- how was multiple testing corrected for?

Reviewer #2: The paper by Belfiore et al describes the measurement of some aortic parameters by using echocardiography in 3 different mouse models of LSD, in particular Pompe Disease, Fabry Disease and Sanfilippo IIIB. Although such evaluation, conducted on each single model and also among the 3 models, showed some statistically significant differences between each model vs. wt mice as well as among the models, and although the Authors state this is the first study on this specific topic in these LSD models, this reviewer finds the study not particularly interesting due to the already available data on the patients affected by the same diseases, who can be much more informative than the mouse models. In my opinion, mouse models are worthy to be tested and used when similar data are nor recoverable from the clinics, also taking into consideration the important differences that might sometimes be registered between the models and the human subjects. In addition, the very limited number of mice examined further reduces the interest of the paper.

In my opinion, the paper is not worthy of publication.

Maybe Authors should have complemented the presented data by pathologically characterizing the cardiac region examined and by providing data obtained from the same mice following treatment, as Authors themselves state in the conclusion of the manuscript.

6. PLOS authors have the option to publish the peer review history of their article (what does this mean?). If published, this will include your full peer review and any attached files.

Reviewer #1: No

Reviewer #2: No

---

## [Author Response · Author response to Decision Letter 0]

1 Mar 2020

Response to reviewers

Reviewer #1

Question n. 1 - Major

This is an interesting manuscript that highlights an aspect of LSDs that is not often considered. I wonder whether the authors can provide a little more data on parameters that are relevant for the aortopathies that they observe? For example cell biological analysis of storage material/lysosomal pathology? Other minor comments are listed below.

Answer n. 1 – Major

Thank you for your comment. As suggested, in the revised version of the manuscript we have provided available data on the histochemical analysis of the KO mice (NAGLU-/- and GAA-/-), describing unique characteristics of the semilunar valve and vessel in these aortic models. We think that this evidence significantly strengthens the paper.

We specified in the text:” H&E staining and PAS sections in GAA-/- mice evidenced an aortic wall with mild disorganization of lamellar units for the presence of many vacuoles containing fine granular or amorphous material in their reduced inter-lamellar space. The aortic valve was not thickened, nor accumulation was found (data not showed). 

H&E staining and PAS sections in in NAGLU-/- mice confirmed the aortic valve defects detected by echocardiography at the time point examined. NAGLU-/- mice exhibited significant aortic cuspid thickening with excess/redundant tissue present in the valves as demonstrated by Alcian blue-PAS staining. By using the H&E staining, NAGLU-/- aortic valve cuspids exhibited a disruption of the normal collagen/proteoglycan boundary interfaces observed in WT animals”

Question n. 2 - Minor

- Literature citations and discussion are incomplete, no mentioning has been made on aortic stiffness in human patients. These are relevant to this manuscript.

Answer n. 2 - Minor

Thank you for your comment. As you suggested, we expanded the literature citations and discussion.

Question n. 3 - Minor

- The introduction is too short in my opinion. The question: 'aortopathies…have not been investigated in pre-clinical models…' is not well explained. Why is this study relevant? It is in my opinion, but please make this more clear and provide a more comprehensive reasoning to introduce the topic

Answer n. 3 - Minor

Thank you for your comment. As suggested, we expanded the introduction, focusing on cardiac and aortic involvement in patients with LSDs, to clarify the aim and the importance of the present study.

Question n. 4 - Minor

- please provide information on the genetic background of the mice and corresponding wild type mice.

Answer n. 4 - Minor

Thank you for your comment. As suggested, we have provided the genetic background of the mice models studied. We specified in the materials and methods as follows: “MPS IIIB knockout mice (NAGLU-/-) were generated by insertion of neomycin resistance gene into exon 6 of NALGU gene on the C57/BL6 background .GAA-/- murine model was created by insertion of plasmid containing both the neomycin-resistance gene and the herpes virus thymidine kinase gene in the pBluescript vector. Six independent cell lines containing the disrupted GAA allele were used to make chimeras that were bred to C57BL/6 females to generate heterozygous mice (F1). Fabry KO mice were generated by a targeted disruption of the α-GAL A gene on the C57Bl/6N back-ground. α-GAL A and wildtype (WT) mice were genotyped by PCR.”

Question n. 5 - Minor

- Please provide the data for the weight of the ko mouse models at analysis

Answer n. 5 - Minor

Thank you for your comment. We have provided the weight of the mouse models investigated. In particular, we specified in the text:” The weight of KO mouse models was not significantly different compared to WT (GLA-/- 29.2 ±2.2, NAGLU-/- 28.9 ±0.7, GAA-/- 28 ±2.0, WT 29 ±1.0).”

Question n. 6 - Minor

- The figures are redundant with the tables and could be removed

Answer n. 6 - Minor

Thank you for your comment. As suggested, we have removed the figures.

Question n. 7 - Minor

- how was multiple testing corrected for?

Answer n. 7 – Minor

Thank you for your comment. To compare the aortic parameters in different mouse models we used t-test or ANOVA (when appropriate). Since the groups were similar, multivariate analysis was not performed. 

Reviewer #2

Question n. 1 - Major

The paper by Belfiore et al describes the measurement of some aortic parameters by using echocardiography in 3 different mouse models of LSD, in particular Pompe Disease, Fabry Disease and Sanfilippo IIIB. Although such evaluation, conducted on each single model and also among the 3 models, showed some statistically significant differences between each model vs. wt mice as well as among the models, and although the Authors state this is the first study on this specific topic in these LSD models, this reviewer finds the study not particularly interesting due to the already available data on the patients affected by the same diseases, who can be much more informative than the mouse models. In my opinion, mouse models are worthy to be tested and used when similar data are nor recoverable from the clinics, also taking into consideration the important differences that might sometimes be registered between the models and the human subjects. In addition, the very limited number of mice examined further reduces the interest of the paper.

In my opinion, the paper is not worthy of publication.

Maybe Authors should have complemented the presented data by pathologically characterizing the cardiac region examined and by providing data obtained from the same mice following treatment, as Authors themselves state in the conclusion of the manuscript.

Answer n. 1 - Major

Thank you for your comment. We think that the paper provides important information in the field of LSDs, since the knowledge about the aortic involvement in these conditions are scant. As suggested, we have provided the histochemical analysis of the available KO mice (NAGLU-/- and GAA-/-) describing unique characteristics of the semilunar valve and vessel in these aortic models. We think that this evidence significantly strengthens the paper. 

Unfortunately, we were not able to investigate the effects of therapy on our models. This aim, although really interesting, would take years and efforts to be completed.

We specified in the text:” H&E staining and PAS sections in GAA-/- mice evidenced an aortic wall with mild disorganization of lamellar units for the presence of many vacuoles containing fine granular or amorphous material in their reduced inter-lamellar space. The aortic valve was not thickened, nor accumulation was found (data not showed). 

H&E staining and PAS sections in in NAGLU-/- mice confirmed the aortic valve defects detected by echocardiography at the time point examined. NAGLU-/- mice exhibited significant aortic cuspid thickening with excess/redundant tissue present in the valves as demonstrated by Alcian blue-PAS staining. By using the H&E staining, NAGLU-/- aortic valve cuspids exhibited a disruption of the normal collagen/proteoglycan boundary interfaces observed in WT animals”

---

## [Decision Letter · Decision Letter 1]

2 Apr 2020

PONE-D-19-29826R1

Aortopathies in mouse models of Pompe, Fabry and Mucopolysaccharidosis lysosomal diseases.

PLOS ONE

Dear Prof., Dr., Limongelli,

Thank you for submitting your manuscript to PLOS ONE. After careful consideration, we feel that it has merit but does not fully meet PLOS ONE’s publication criteria as it currently stands. Therefore, we invite you to submit a revised version of the manuscript that addresses the points raised during the review process.

The reviewers still have concerns and indeed, not all points raised by them in the first revision were addressed. The main concern of the reviewers, besides the  small number of animals used in the study, is that each disease group must be compared to wild types from each respective backgrounds. This comparison must be done to make the manuscript acceptable.

In addition, all disease animals were all evacuated at 12 month although as pointed by the reviewer they may present different disease course. The reason for evaluating all groups at this time must be provided with a discussion of the disease course in each group

the quality of the figures should be improved

We would appreciate receiving your revised manuscript by May 17 2020 11:59PM. To enhance the reproducibility of your results, we recommend that if applicable you deposit your laboratory protocols in protocols.io, where a protocol can be assigned its own identifier (DOI) such that it can be cited independently in the future. For instructions see: http://journals.plos.org/plosone/s/submission-guidelines#loc-laboratory-protocols

We look forward to receiving your revised manuscript.

Kind regards,

Andrea Dardis, Ph.D.

Academic Editor

PLOS ONE

Reviewers' comments:

Reviewer's Responses to Questions

**Comments to the Author**

1. If the authors have adequately addressed your comments raised in a previous round of review and you feel that this manuscript is now acceptable for publication, you may indicate that here to bypass the “Comments to the Author” section, enter your conflict of interest statement in the “Confidential to Editor” section, and submit your "Accept" recommendation.

Reviewer #1: (No Response)

Reviewer #2: (No Response)

Reviewer #3: (No Response)

2. Is the manuscript technically sound, and do the data support the conclusions?

Reviewer #1: Partly

Reviewer #2: Partly

Reviewer #3: (No Response)

3. Has the statistical analysis been performed appropriately and rigorously? 

Reviewer #1: Yes

Reviewer #2: I Don't Know

Reviewer #3: (No Response)

4. Have the authors made all data underlying the findings in their manuscript fully available?

Reviewer #1: Yes

Reviewer #2: Yes

Reviewer #3: (No Response)

5. Is the manuscript presented in an intelligible fashion and written in standard English?

Reviewer #1: No

Reviewer #2: Yes

Reviewer #3: (No Response)

6. Review Comments to the Author

Reviewer #1: The authors have significantly improved the manuscript. The histochemical data are partially informative. In the legend to Fig1, no A or B has been described. The quality of Figures 2 (out of focus) and 3 (no details, unclear phenotype, very small and damaged preparations) is not so good. Do the authors have better pictures? Otherwise I would consider removing these.

The English needs to be revised.

Reviewer #2: 1) Although the manuscript was slightly revised, the Authors did not explain why they evaluated a so small number of animals. In addition, Authors did not specify in each Table or Figure the number of animals analyzed in each analysis and did not indicate which Wild Type mice were analyzed. It is known that WT mice deriving from different colonies may not share similar features. Thus, each model should include for comparison wild type mice deriving from the same colony, whereas from the manuscript it seems that Authors used a general group of wild type mice with no specific reference to each colony.

2) All along the manuscript MPS IIIB is called MPS, which is a misleading definition, since features of MPS IIIB may be not extendable to all MPS. We know for example that as for other cardiologic features, as valvulopathies or others, not all MPSs present the same degree of involvement or even the same presence/absence in the patients along their pathological progression, so to ascertain that the described features, related to aortopathies, were only identified in the MPS IIIB mouse model is mandatory. This should be described also in the discussion in more details.

3) Why did the Authors analyze animals at 12 months of age? Authors should explain why they decided to evaluate all animals at the same age while different models may show pathological signs/symptoms at different ages. Very likely at 12 months of age all models present a very pathological phenotype. Which was the disease progression in the different models?

Reviewer #3: In this work, the Belfiore et al have investigated the presence and characteristics of aortopathy related to three lysosomal storage disorder models Pompe Disease (PD), Fabry disease (FD) and mucopolysaccharidoses (MPS). In a series of experiments the authors have reported differences and in the revised version the authors have supplied some additional experimental evidence to answer one of the reviewers.

The second reviewer who recommended rejection based this decision on the small numer of animals used in the experiments and not providing data following ERT treatment.

Although it is understandable that providing data following treatment might take too long it is not clear to this reviewer whether all the different mouse strains were tested with wild-types from each respective background. If this has not been done then it would be important to include it in a eventually revised version of the manuscript

7. PLOS authors have the option to publish the peer review history of their article (what does this mean?). If published, this will include your full peer review and any attached files.

Reviewer #1: No

Reviewer #2: No

Reviewer #3: No

---

## [Author Response · Author response to Decision Letter 1]

11 Apr 2020

Response to Reviewers.

Reviewer #1:

The authors have significantly improved the manuscript. The histochemical data are partially informative. In the legend to Fig1, no A or B has been described. The quality of Figures 2 (out of focus) and 3 (no details, unclear phenotype, very small and damaged preparations) is not so good. Do the authors have better pictures? Otherwise I would consider removing these. The English needs to be revised.

Thank you for your comment. Pictures have been modified and improved in their quality. English have been revised all over the text.

Reviewer #2:

1) Although the manuscript was slightly revised, the Authors did not explain why they evaluated a so small number of animals. In addition, Authors did not specify in each Table or Figure the number of animals analyzed in each analysis and did not indicate which Wild Type mice were analyzed.

Thank you for your comment. This study was essentially designed to describe the aortic phenotype in different mouse models of lysosomal storage disease. To this aim, we believe that sample size was efficacious to verify our hypothesis. Moreover, according to your suggestion, the number of animals analyzed in each analysis was specified in each Table or Figure.

It is known that WT mice deriving from different colonies may not share similar features. Thus, each model should include for comparison wild type mice deriving from the same colony, whereas from the manuscript it seems that Authors used a general group of wild type mice with no specific reference to each colony. 

Wild type mice were absolutely derived from the same colony of their correspondent LSD mouse model. We corrected the text all over indicating that phenotypic comparisons have been performed against wild type mice deriving from the same colony of their correspondent KO mice.

2) All along the manuscript MPS IIIB is called MPS, which is a misleading definition, since features of MPS IIIB may be not extendable to all MPS. We know for example that as for other cardiologic features, as valvulopathies or others, not all MPSs present the same degree of involvement or even the same presence/absence in the patients along their pathological progression, so to ascertain that the described features, related to aortopathies, were only identified in the MPS IIIB mouse model is mandatory. This should be described also in the discussion in more details.

In agreement with reviewer suggestion, we modified all along the manuscript the meaning of MPS by specifying that our model is referred to the specific MPS IIIB disease. We specified this also in the discussion section.

3) Why did the Authors analyze animals at 12 months of age? Authors should explain why they decided to evaluate all animals at the same age while different models may show pathological signs/symptoms at different ages. Very likely at 12 months of age all models present a very pathological phenotype. Which was the disease progression in the different models?

Thank you for your important comment. Your assumption is right, since at 12 months of age all models are very likely to show pathological phenotype (including a progressive phenotype as aortopathy). The main aim of this study was to describe the aortic phenotype in different mouse models of lysosomal storage disease. On the other hand, this pilot study was not designed to describe aortic progression in different models and to test different therapies (i.e. ERT, chaperone, genetic therapy, etc). Therefore, we do not have data on different aortic parameters progression, and we stated this in the study limitation. 

Reviewer #3:

In this work, the Belfiore et al have investigated the presence and characteristics of aortopathy related to three lysosomal storage disorder models Pompe Disease (PD), Fabry disease (FD) and mucopolysaccharidoses (MPS). In a series of experiments the authors have reported differences and in the revised version the authors have supplied some additional experimental evidence to answer one of the reviewers.

The second reviewer who recommended rejection based this decision on the small number of animals used in the experiments and not providing data following ERT treatment.

Although it is understandable that providing data following treatment might take too long it is not clear to this reviewer whether all the different mouse strains were tested with wild-types from each respective background. If this has not been done then it would be important to include it in a eventually revised version of the manuscript.

Thank you for your comment. Wild type mice were absolutely derived from the same colony of their correspondent LSD mouse model. We corrected the text all over indicating that phenotypic comparisons have been performed against wild type mice deriving from the same colony of their correspondent KO mice.

---

## [Decision Letter · Decision Letter 2]

28 Apr 2020

Aortopathies in mouse models of Pompe, Fabry and Mucopolysaccharidosis IIIB lysosomal storage diseases.

PONE-D-19-29826R2

Dear Dr. Limongelli,

We are pleased to inform you that your manuscript has been judged scientifically suitable for publication and will be formally accepted for publication once it complies with all outstanding technical requirements.

With kind regards,

Andrea Dardis, Ph.D.

Academic Editor

PLOS ONE

Additional Editor Comments (optional):

Reviewers' comments:

Reviewer's Responses to Questions

**Comments to the Author**

1. If the authors have adequately addressed your comments raised in a previous round of review and you feel that this manuscript is now acceptable for publication, you may indicate that here to bypass the “Comments to the Author” section, enter your conflict of interest statement in the “Confidential to Editor” section, and submit your "Accept" recommendation.

Reviewer #1: All comments have been addressed

Reviewer #3: All comments have been addressed

2. Is the manuscript technically sound, and do the data support the conclusions?

Reviewer #1: Yes

Reviewer #3: Yes

3. Has the statistical analysis been performed appropriately and rigorously? 

Reviewer #1: Yes

Reviewer #3: Yes

4. Have the authors made all data underlying the findings in their manuscript fully available?

Reviewer #1: Yes

Reviewer #3: Yes

5. Is the manuscript presented in an intelligible fashion and written in standard English?

Reviewer #1: Yes

Reviewer #3: Yes

6. Review Comments to the Author

Reviewer #1: The authors have addressed my comments and I recommend publication. This is original work that is of value for the scientific community.

Reviewer #3: Authors have clarified my previous observation regarding the type of controls that have been used. The manuscript has also been improved in the remaining areas highlighted by the other reviewer

7. PLOS authors have the option to publish the peer review history of their article (what does this mean?). If published, this will include your full peer review and any attached files.

Reviewer #1: No

Reviewer #3: No

---

## [Editor Report · Acceptance letter]

1 May 2020

PONE-D-19-29826R2 

Aortopathies in mouse models of Pompe, Fabry and Mucopolysaccharidosis IIIB lysosomal storage diseases. 

Dear Dr. Limongelli:

I am pleased to inform you that your manuscript has been deemed suitable for publication in PLOS ONE. Congratulations! Your manuscript is now with our production department. 

With kind regards,

on behalf of

Dr. Andrea Dardis 

Academic Editor

PLOS ONE